# Influence of Gamma Irradiation on Electric Cables Models: Study of Additive Effects by Mid-Infrared Spectroscopy

**DOI:** 10.3390/polym13091451

**Published:** 2021-04-30

**Authors:** Astrid Maléchaux, Juliette Colombani, Sandrine Amat, Sylvain R. A. Marque, Nathalie Dupuy

**Affiliations:** 1Aix Marseille University, Avignon Université, CNRS, IRD, IMBE, 13013 Marseille, France; amalechaux@ondalys.fr (A.M.); sandrine.amat@univ-amu.fr (S.A.); 2IRSN, CEN Cadarache, BP3, 13115 St Paul lez Durance, France; juliette.colombani@irsn.fr; 3Institut de Chimie Radicalaire, Aix Marseille Universitè, CNRS, ICR, Case 551, 13397 Marseille, France; sylvain.marque@univ-amu.fr

**Keywords:** nuclear power plant cables, gamma irradiation, polymer ageing, antioxidant, infrared spectroscopy, principal component analysis

## Abstract

Cables, especially their insulation and jacket materials made of polymers, are vulnerable to ageing degradation during normal operation. However, they must remain functional for the entire life of a nuclear power plant, or even in the event of an accident for cables with a safety requirement. This study focuses on models of crosslinked polyethylene (XLPE)-based insulation of cables and deals with the structure modification and the behavior of XLPE for nuclear applications due to the effect of additives. Various additives are added to the polymer formulation to evaluate their impact on ageing. The samples are irradiated at room temperature by several gamma doses, up to 374 kGy, with two dose rates (40 Gy/h and 300 Gy/h) and compared with a non-irradiated sample used as reference. To understand the impact of gamma irradiation on the materials, the principal component analysis (PCA) method is applied on spectra recorded through attenuated total reflectance–Fourier transform infrared (ATR-FTIR) spectroscopy. The results highlight the effects of ageing depending on the dose rate and on the formulation of the materials, with the identification of different degradation products. A curve resolution study compares the effects of different additives on polymer oxidation and shows that the low dose rate leads to a higher degradation than the high dose rate.

## 1. Introduction

The effects of radiation on polymeric materials are a topic of concern in a range of industries including sterilization [1] and nuclear power plants (NPPs) [2]. In this last frame, the prediction of polymers′ long-term durability is crucial for nuclear safety, particularly for the polymer comprising the electrical cable connecting important safety systems. Indeed, these electric cables are long-life equipment that is often difficult, if not impossible, to replace during the operation life of a NPP, and are exposed, in normal operating conditions of a NPP, to a dose rate of about 0.1 Gy/h at a temperature below 50 °C and a humidity level of 70%. Moreover, new reactors are being constructed or planned over coming years, and in these cases, it will also be very useful for an effective cable lifetime management to choose appropriate cable materials [3]. Therefore, studies on the ageing of cables under radiation are of major importance to predict their lifetime and durability. This is the scope of the European Union (EU) project called TeaM Cables which aims at developing cable ageing models and algorithms, based on multi-scale studies of their degradation under different conditions: gamma irradiation and/or thermal ageing [4]. The thermal degradation [5] and gamma radiation [6,7] of polyethylene has been intensively studied. In this work, only degradation under gamma irradiation is studied, all experiments taking place at room temperature.

The degradation could be observed in terms of formation of radicals [8], small molecules, inter- and intra-chain bonding, unsaturations, colorings [9] and formation of oxidized molecules [10,11], resulting in crosslinking and/or chain scission [12,13] which could alter the mechanical properties of the material [14,15]. The degradation of additives is also commonly observed [16,17]. These modifications depend on polymer formulation, gamma dose, dose rates and ageing post irradiation [18,19]. Dose rates induced changes in degradation pathways. For example, low dose rate is known to result in changes caused by diffusion-limited oxidation [20].

This study focuses on models of crosslinked polyethylene (XLPE)-based insulation of cables and deals with the structure modification and the behavior of XLPE for nuclear applications because of additives. To understand the impact of gamma irradiation at room temperature on polymers, several model samples containing either none, one, or a mixture of the additives Irganox 1076 (antioxidant), Irganox PS802 (heat stabilizer) and alumina trihydrate (ATH, flame-retardant), were manufactured and analyzed. Material responses were investigated through mid-infrared spectroscopy which appeared to be very sensitive to changes in filler composition. The spectroscopic data were then analyzed using the principal component analysis (PCA) method to explain the ageing according to the formulation of the materials. This global analytical technique is nevertheless restrictive due to the limited identification of the main components. The infrared spectra are the result of the additivity of all the spectral bands of all the molecules present in the sample (polymer, additives, and degradation products). Owing to the complex chemical composition of film, a curve resolution method for self-modeling analysis named SIMPLISMA is used. This mathematical method allows pure spectra and associates the “contributions” of different substances present in polymers without prior information of their existence needing to be obtained.

Since infrared spectroscopy allows us to follow the evolution of polymers on the surface, differential scanning calorimetry (DSC) measurements have been carried out to follow the evolution of the bulk materials.

## 2. Materials and Methods

### 2.1. Samples

The tape samples were manufactured with a dimension of 50 mm width, 0.5 mm thickness and 80 mm length. The small thickness of the samples aims to avoid as much as possible the limitation of the diffusion of oxygen in the material. The material can be therefore considered as homogeneously degraded over the major part of its thickness. Six different formulations were obtained, starting with only silane XLPE. The XLPE used linear low-density polyethylene as the base polymer, vinyltrimethoxysilane (VTMO) as crosslinking agent and dicumyl peroxide (DCP) as the initiator. The second material consisted of 99% *w*/*w* of XLPE and 1% *w*/*w* of the antioxidant Irganox 1076. The third material consisted of 99% *w*/*w* of XLPE and 1% *w*/*w* of the heat stabilizer Irganox PS802. The chemical structures of these two additives are detailed in Figure 1. Then, a formulation used a combination of the two Irganox additives (98% *w*/*w* XLPE + 1% *w*/*w* Irganox 1076 + 1% *w*/*w* Irganox PS802). The last one was made of 67% *w*/*w* of XLPE and 33% *w*/*w* of the flame-retardant ATH. Finally, a formulation combined all three additives (65.8% *w*/*w* XLPE + 32.8% *w*/*w* ATH + 0.7% *w*/*w* Irganox 1076 + 0.7% *w*/*w* Irganox PS802).

### 2.2. Mid-Infrared Spectroscopy

Mid-infrared (MIR) spectra were recorded with a Nicolet IS50 FT-IR (Fourier transform infrared) (THERMO FISHER SCIENTIFIC, Villebon sur Yvette, France) spectrometer using attenuated total reflectance (ATR) mode. The contact between analyzed sample and the diamond of the ATR accessory was ensured by screwing a clamp device. The MIR spectrometer was situated in an air-conditioned room (21 °C) and air was taken as reference for the background spectrum before each sample. Between each spectrum, the ATR accessory was cleaned with ethanol solution, enabling us to dry the ATR. Cleanliness was verified by collecting a background spectrum and comparing it to the previous background spectrum. A total of 100 scans were co-added to generate a spectrum with a 4 cm^−1^ spectral resolution on the 670–4000 cm^−1^ spectral range. Three spectra of each sample were recorded on three different parts of the sample and on different days. All these spectra were directly used for chemometric analysis to be able to distinguish variance of the triplicates from variances of potential chemical modifications.

### 2.3. Differential Scanning Calorimetry (DSC) under O_2_ (Oxidative Induction Time (OIT) Measurements)

The principle of oxidative induction time (OIT) measurement is based on DSC analysis, it measures the difference in heat flow (in mW) between a sample and an inert reference during an imposed thermal change, under a controlled atmosphere. During OIT measurements, the time required to induce the oxidation process under pure O_2_ flow at a given temperature (isotherm) is measured. The induction of the oxidation process is characterized by an increase of the heat flow which enables the OIT to be determined. For stabilized polymers, OIT measurements can be used to give an assessment of the level of stabilization of the material, as OIT is the time required to consume all the stabilizers.

OIT measurements were performed under a pure O_2_ flow at 195 +/− 15 °C using a Setaram Instrument DSC 131 Evo calorimeter (Caluire-et-Cuire, France). The temperature was chosen to have the most suitable duration (maximum duration or greater than 150 min) for the experiments carried out on the unaged samples. Samples with a mass ranged between 5 +/− 3 mg were introduced in opened standard aluminium pans. These samples were first heated, under pure N_2_ flow of 50 mL·min^−1^, from room temperature to 195 +/− 15 °C at a heating rate of 10 °C·min^−1^. After an isotherm segment of 5 min under N_2_, the gas supply was switched from N_2_ to O_2_ while maintaining the same gas flow (50 mL·min^−1^) to access the oxidation induction time (OIT). For each sample, at least three measurements were performed. OIT was then determined with the tangent method, corresponding to the duration between the introduction of O_2_ flow in the DSC cavity and the onset of the oxidation exotherm.

### 2.4. Irradiation

Accelerated radiolytic ageing (RA) was conducted at room temperature (RT) with medium (40 Gy/h) and high (300 Gy/h) dose rates with periodical withdrawals, resulting in the total integrated doses presented in Table 1. As explained above, the material is irradiated and degraded homogeneously over the entire thickness of the material.

The accelerated irradiation at medium dose rate (40 Gy/h) was conducted at Panoza (cobalt irradiator) facility with a room temperature around 45 °C, while the accelerated irradiation at high dose rate (300 Gy/h) was conducted at the Prazdroj facility with room temperature around 30 °C. Both facilities are in the Nuclear Research Institute (Rež, Czech Republic). Tape samples were drilled and fixed to perforated stainless steel plates. The plates were then placed inside a metal cylinder with a diameter of 90 cm (Panoza) or 80 cm (Prazdroj) and height of 50 cm. The plates were flipped over every three weeks to ensure the homogeneity of the radiation field.

### 2.5. Chemometrics

#### 2.5.1. Curve Resolution Method (SIMPLISMA)

The method used for self-modeling mixture analysis (the SIMPLISMA method) is described in the literature [21,22,23,24,25,26]. This method is used for self-modeling mixture analysis by resolving mixture data into pure component spectra and concentration. When overlapping spectral features are present in spectroscopic data, this tool is unable to resolve broad spectral components and separate spectral absorption bands characterizing one component. Its concept is based on the determination of pure variables (e.g., a wavenumber in spectroscopic terms) that have received the contributions from only one component and an optimization using least-square method.

#### 2.5.2. Principal Component Analysis (PCA)

Principal component analysis (PCA) is a tool commonly used in chemometrics and was described in a previous study [10]. Each principal component (PC) is built to maximize the variance extracted from the remaining data. The projection of the scores in the space defined by the PCs gives an overview of the similarities and differences between the samples, while the loadings indicate which variables bring more information to each PC [27]. In this study, we considered for each model only the PCs containing more than 15% of the variance. Thus, in some cases only the first PC was relevant, and the scores could be represented as a bar chart. In other cases, the first two PCs were relevant, and the scores could be represented as a scatter plot.

#### 2.5.3. Data Treatments

A unit vector normalization was applied to compensate for additive and/or multiplicative effects. The standard normal variate (SNV) method allows a normalization of the spectra that consists in subtracting each spectrum by its own mean and dividing it by its own standard deviation. After SNV, each spectrum will have a mean of 0 and a standard deviation of 1. The upward shift of the baseline in the 995–860 cm^−1^ spectral range is corrected with Savitzky Golay derivative pre-processing [10].

#### 2.5.4. Software

Chemometrics analysis was performed using Unscrambler X V.10.3 (CAMO/Software, Oslo, Norway). For SIMPLISMA analysis, chemometrics pre-treatments (baseline correction) were applied using Unscrambler^®^X (version 10.4, CAMO Software). The pre-treated matrices were then imported to Matlab^®^ (version 7.8 R2009a, MathWorks, Natick, MA, USA) for data analysis.

## 3. Results

### 3.1. Characterization of Unaged Samples by Mid-Infrared Spectroscopy Subsection

The mid-infrared spectra of the non-irradiated samples are presented in Figure 2. The infrared (IR) assignments of the bands are provided in Table 2 according to the literature [10,28,29]. Peaks at 2914 and 2848 cm^−1^ are attributed to the antisymmetric and symmetric stretching of –CH_2_– groups. Peaks included between 1470 and 1460 cm^−1^ are ascribed to the deformation of –CH_2_– and –CH_3_– groups. Peaks at 720–730 cm^−1^ are characteristic of long chains of –CH_2_– in XLPE.

For the PCA, only the 1800–670 cm^−1^ spectral zone was retained. The PCA was performed on 30 samples corresponding to non-irradiated samples and irradiated samples at two dose rates (300 Gy/h and 40 Gy/h). For each dose rate, 5 irradiation doses were applied 67, 145, 220, 298 and 374 kGy for the 300 Gy/h dose rate and 67, 134, 202, 269 and 336 kGy at 40 Gy/h dose rate. Spectra recorded in triplicate were adjusted with baseline correction and SNV normalization.

### 3.2. Principal Component Analysis (PCA) on Fourier Transform Infrared (FTIR) Spectra Impact of Irradiation

#### 3.2.1. Crosslinked Polyethylene (XLPE) without Additive

The first principal component (PC1) represents 86% of the total variance included in the spectra. Changes are correlated to the dose rate and to the total absorbed dose. Reading the scores associated with the first component shows that the impact of gamma irradiation is less important for the 300 Gy/h dose rate than for the 40 Gy/h dose rate (Figure 3). The first dose associated with each dose rate is identical (67 kGy) and the scores associated with the 300 Gy/h dose rate are higher compared to the 40 Gy/h dose rate. For each dose rate the scores decrease with the dose. The loading associated to the first principal component shows that when the absorbed dose increases, there is an increase in the intensity of the peak associated to carbonyl function (the negative loading is correlated with the negative scores so the lower the scores, the more important the band). The band associated with the carbonyl function is very broad and includes carboxylic acid (1700–1740 cm^−1^), ketones (1705–1725 cm^−1^), aldehydes (1720–1740 cm^−1^) and/or ester (1725–1750 cm^−1^) [30]. No signal is observed at 1780 cm^−1^, corresponding to the lactone. The complex band around 1170 cm^−1^ is attributed to (δ–O–H, ν–C–O) combination (ester or acid forms). In the positive part the two bands at 1467 and 715 cm^−1^ are attributed to the XLPE polymer. As a matter of fact, the signal associated with the non-degraded polymer is more important at low dose than at high dose. Thus, the XLPE polymer without additive is clearly degraded by gamma irradiation, and the degradation increases with the integrated dose received by the material. Moreover, the stronger effect of the 40 Gy/h dose rate compared to the 300 Gy/h dose rate could be explained by the longer exposure time required to reach a similar total dose at the lower dose rate. At 300 Gy/h dissolved oxygen is used up faster than at 40 Gy/h and we could suppose it cannot be replaced by atmospheric oxygen by permeation through the film. At 40 Gy/h the the oxidation will proceed further into the sample [19]. Indeed, a longer exposure time enhances the possibility of the reaction of radicals with oxygen inducing the oxidative degradation of the material (formation of carboxylic acid, ketones, aldehydes and so on, as pointed out above). The effect of temperature cannot be ruled out, because the temperature for irradiations at 40 Gy/h is 40 °C while that for irradiations at 300 Gy/h is 300 °C. The difference is small, but it could play a role [31].

Figure 4 concerns the unsaturations zone [995–860 cm^−1^]. For a better interpretation and to remove the variations of the baseline, spectra are presented plotted in first derivative. On the overlay shown in Figure 4, a band at 970 cm^−1^ which corresponds to the first derivative of 964 cm^−1^ band (–R_1_–CH=CH–R_2_, trans vinylene groups), appears with gamma irradiation.

Peaks at 914 cm^−1^ (–R–CH=CH_2_, vinyl end groups) ^1^ which correspond to the first derivative of 908 cm^−1^ decrease slightly, and at 890 cm^−1^ (R_1_R_2_C=CH_2_, vinylidene groups) does not vary. These observations agree with literature [6]. The formation of unsaturated groups is monotonic with the absorbed dose. The irradiation generates trans-vinylene groups, slightly degrades the vinyl end groups, and does not alter the vinylidene groups [6,7,11].

#### 3.2.2. XLPE (99% *w*/*w*) with Irganox 1076 (1% *w*/*w*)

The first principal component (PC1) represents 89% of the total variance included in the spectra. Changes are first correlated to the dose rate. Reading the scores associated with the first component shows that the impact of gamma irradiation is also in this case less important for the 300 Gy/h dose rate than for the 40 Gy/h dose rate (Figure 4). For the dose rate 40 Gy/h the impact of gamma irradiation is not correlated to the dose. As a matter of fact, the associated scores are not very different for the five doses. For the dose rate 300 Gy/h the impact of gamma irradiation is correlated to the dose, even if the impact of the modification is not homogenous for the samples irradiated at 135 kGy and at 336 kGy. However, this heterogeneity could in fact be due to the distribution of the additive in the initial sample composition since the scores of the non-irradiated samples already present some variability. The loadings associated with the first principal component show that the positive bands could be attributed to Irganox 1076 (Table 2). In the negative part of the loadings the two bands at 1471 and 715 cm^−1^ are attributed to the XLPE polymer (Table 2). These results could be interpreted in two different ways. On the one hand, the consumption of Irganox 1076 increases with the dose received at 300 Gy/h, whereas a strong consumption occurs at 40 Gy/h regardless of the dose. The polymer is not degraded whatever the dose rate and the dose, since there is no oxidation band contrary to what was observed for XLPE without additive. On the other hand, physical loss of additives can occur by blooming effect. As a matter of fact, Xu et al. [29] showed that the exudation and blooming of Irganox 1076 on to the surface of polyethylene-based materials appear for concentration superior to 0.9 wt% Irganox 1076 in the XLPE matrix [29]. For the authors, this value corresponds to the critical concentration of antioxidants above which a phase separation is detected. However, as the concentration of Irganox 1076 was to 1wt% in the polymer studied in this part, it is difficult to imagine that the blooming effect could be the only phenomenon to occur. Thus, a combination of physical loss and chemical degradation of the additive should be considered.

The comparison of non-irradiated spectra and spectra after gamma irradiation (Figure 5) shows the consumption of Irganox 1076. This phenolic antioxidant inhibits the propagation of free radicals that are stabilized by its aromatic cycle [31]. The degradation pathways of this additive are complex and numerous as explained by Dorey and et al. [16]. The many degradation products result in a spectrum with no clearly identifiable bands except for those of the XLPE polymer, which seems to be protected by the additive.

#### 3.2.3. XLPE (99% *w*/*w*) with Irganox PS802 (1% *w*/*w*)

The first principal component (PC1) represents 83% of the total variance included on the spectra. Changes are first correlated to the dose rate. Reading the scores associated with the first component shows that the impact of gamma irradiation is also in this case less important for the 300 Gy/h dose rate than for the 40 Gy/h dose rate (Figure 6). For the dose rate 40 Gy/h the degradation is less important at 67 kGy but then the impact of gamma irradiation no longer varies with the dose in the 145–374 kGy range. For the 300 Gy/h dose rate there is no degradation of the polymer until 336 kGy. The loadings associated with the first principal component show that the positive bands at 1311, 1263 and 1130 cm^−1^ could not be attributed to Irganox PS802 (Table 2). Some bands associated with the loadings present important shifts compared to the non-irradiated polymer (1733 cm^−1^ was transformed in two bands at 1737 cm^−1^ and 1720 cm^−1^, 1467 cm^−1^ shifted to 1423 cm^−1^, 1361 cm^−1^ shifted to 1363 cm^−1^, and 1180 cm^−1^ shifted to 1187 cm^−1^). In the negative part of the loading the two bands at 1467 and 715 cm^−1^ are attributed to the XLPE polymer according to the Table 2. These results show that the Irganox PS802 is consumed according to the dose at 300 Gy/h, and is consumed independently of the dose at 40 Gy/h.

This is confirmed by the observation of FTIR bands (Figure 5). There is not much change in bands attributed to the polymer but some shifts in the bands attributed to the additive and the appearance of new bands that can be attributed to oxidation products of the additive. Irganox PS802 being a thioether, its role is to inhibit the formation of hydroperoxides [32], and it is mostly used to protect the polymer against thermal ageing. In the case of thermal degradation, Xu et al. [33] estimate that physical loss by evaporation is the main ageing process compared to chemical consumption. However, in the case of radiation degradation studied here the FTIR spectra indicate a strong effect of chemical consumption of the additive. The most likely oxidation product is the 3,3′-thiodipropionic acid, the characteristic bands of this product are at 1720 cm^−1^ ν–C=O acid (stretching); 1363 cm^−1^ ω–CH_2_–S (wagging), 1311 cm^−1^ to 1263 cm^−1^, δ O–H acid (deformation), 1234 cm^−1^ ν C–O (asymmetric stretching), 1187 cm^−1^ δ–O–H, ν C–O (combination), 1130 cm^−1^ ν C–O acid (stretching) [28,33] and they are well in line with the information shown in this loading. The degradation of 3,3′-thiodipropionic acid is tentatively described in Figure 7. Indeed, under γ-irradiation, a radical cation is generated on the ester moiety of Irganox PS802 which rearranges into protonated carbonyl moiety and alkyl radical via 1,5-H transfer. After the loss of proton, the alkyl radicals collapse into heavy alkene and carbonyloxyl radical. The latter abstracts an H-atom in its environment to generate a carboxylic moiety. The same radical sequence occurs with the other ester moiety of this carboxylic acid to yield the 3,3′-thiodipropionic acid. However, the degradation of the additive does not go along with a decrease in the bands assigned to it. This can be due to the migration of the additive towards the surface of the sample as explained by Xu [20].

#### 3.2.4. XLPE (98% *w*/*w*) with Irganox 1076 (1% *w*/*w*) and Irganox PS802 (1% *w*/*w*)

The first principal component (PC1) represents 69% of the total variance included on the spectra. Once again, the distribution of the additives appears to be heterogeneous even in the non-irradiated samples. Changes are first correlated to the dose rate. Reading the scores associated with the first component shows that the impact of gamma irradiation is also in this case less important for the 300 Gy/h dose rate than for the 40 Gy/h dose rate (Figure 8). For the 40 Gy/h dose rate, the degradation starts at 67 kGy and strongly increases with the dose, whereas for the 300 Gy/h dose rate no significant effect is observed before 134 kGy and the impact of irradiation remains similar in the 134–336 kGy range. This material is expected to benefit from the combined effects of the two additives additives, but the loadings associated with PC1 are very close to those obtained with only Irganox PS802, indicating a preponderance of the degradation of this additive over Irganox 1076. However, the ratio between the scores at 1737 and 1718 cm^−1^ is different than for XLPE with only Irganox PS802. Here, the intensity at 1718 cm^−1^ is lower which suggests that the additive is less degraded into 3,3′-thiodipropionic acid.

The FTIR bands (Figure 8) confirm similar behavior between this material and the one with only Irganox PS802 but with less intense bands related to the oxidation product of the additive in when Irganox 1076 is also present, especially with the 300 Gy/h dose rate. There is not much change in the FTIR bands attributed to the polymer, showing that it is again protected from radiolytic ageing.

#### 3.2.5. XLPE (67% *w*/*w*) with ATH (33% *w*/*w*)

As shown on the PCA scores (Figure 9), the first principal component (PC1) represents 60% of the total variance included in the spectra and the second principal component (PC2) corresponds to 36% of the same variance. According to the first principal component, changes are first correlated to the dose rate. Reading the scores associated with the first and second components shows that the impact of gamma irradiation is not visible until 145 kGy for the 300 Gy/h dose rate and until 269 kGy for the 40 Gy/h dose rate (Figure 9).

The samples are separated into two groups on the PC2; the first one corresponds to the 300 Gy/h dose rate and the second one to the 40 Gy/h dose rate. The impact of gamma irradiation is related to the degradation of the polymer characterized by the increase of the carbonyl band ν–C=O at 1712 cm^−1^ and the appearance of the vibration ν C–O at 1170 cm^−1^. These two infrared bands are characteristic of the acid form. The negative part of the PC1 at 1014 cm^−1^ is characteristic of ATH additive in non-degraded polymers according to the Table 2. The second principal component expresses the difference between the two dose rates. The positive part corresponding to the dose rate at 40 Gy/h is characteristic of the spectrum of the additive ATH (1014 cm^−1^, 964 cm^−1^, 792 cm^−1^) except for the band at 1712 cm^−1^ corresponding to the acid form, whereas the negative part corresponding to the dose rate at 300 Gy/h is characteristic of the polymer. Thus, the ATH additive does not efficiently protect the XLPE polymer against radiolytic ageing, especially at the 40 Gy/h dose rate.

#### 3.2.6. XLPE (65.8% *w*/*w*) with ATH (32.8% *w*/*w*), Irganox 1076 (0.7% *w*/*w*) and Irganox PS802 (0.7% *w*/*w*)

The first principal component (PC1) represents 62% of the total variance included in the spectra and the second principal component (PC2) corresponds to 21% of the same variance. The scores show a strong heterogeneity in the composition of the non-irradiated samples, due to the distribution of the three additives. This makes it difficult to detect the effect of the integrated dose, but changes are still correlated to the dose rate. Reading the scores associated with the first component shows that the impact of gamma irradiation is not visible until 298 kGy for the 40 Gy/h dose rate (Figure 10). For the 300 Gy/h dose rate, interpretation is more difficult since low doses and high dose are classified in the same space whereas the middle dose seems to be less degraded. The samples are separated into two groups on the PC2; the first one corresponds to the 300 Gy/h dose rate and the second one to the 40 Gy/h dose rate. For the first principal component, the impact of gamma irradiation is associated with a decrease of 1020 cm^−1^ band principally attributed to ATH additive (according to Table 2). The degradation of ATH brings out the bands of the other additives, Irganox 1076 and Irganox PS802 at 1731, 1410, 1357, 1261, 1240, 1180, and 1132 cm^−1^, for both dose rates. For the second principal component that characterizes the difference between the 2 dose rates, the positive part is attributed to the dose rate at 40 Gy/h. The characteristic bands are attributed to the polymer (1469 cm^−1^), to the additives Irganox 1076 (1434 cm^−1^) and Irganox PS802 (1355 cm^−1^, 1241 cm^−1^ and 1176 cm^−1^). The negative part is attributed to the dose rate at 300 Gy/h with bands attributed to the degradation product of Irganox PS802 (1714 cm^−1^, 1263 cm^−1^ and 1188 cm^−1^). Thus, the XLPE polymer appears to be protected by the combination of the three additives.

The FTIR spectra (Figure 10) show the appearance of bands attributed to the degradation product of Irganox PS802, especially for the dose rate at 40 Gy/h. The effect on the other additives is not easily visible and some variations of the absorption could also be caused by a migration of the additives towards the surface of the material [29].

### 3.3. Curve Resolution Results

Each FTIR experiment was realized on the 6 formulations in triplicate at all the irradiation doses and dose rates so the SIMPLISMA treatment was applied to all spectra (39 spectra/formulation = 234 spectra, see Table 3).

This treatment gave 6 pure variables and their corresponding extracted concentration profiles. A pure variable is defined as a variable whose intensity mainly results from one of the components in the mixture under consideration [22,23,24,25]. Concentration profiles do not represent a real concentration but the contribution of each pure component in each spectrum.

The first pure spectrum (Figure 11a) corresponds to the ATH species, the characteristic infrared bands are found at 1020 cm^−1^, 966 cm^−1^, 796 cm^−1^ and 735 cm^−1^ according to the identifications given in Table 2. The concentration profile (Figure 11b) shows high contribution between spectra 157 to 234 corresponding to the two last formulations (XLPE + ATH and XLPE + ATH + Irg1076 + IrgPS802). No degradation of the ATH is highlighted regardless of dose and dose rate, since no decrease in this contribution is observed (excepted that due to the decrease in the amount of ATH in the last formulation).

The second pure spectrum (Figure 12a) corresponds to Irganox PS802 spectrum, the characteristic infrared band are found at 1733 cm^−1^, 1463 cm^−1^, 1355 cm^−1^ 1240 cm^−1^, 1180 cm^−1^ and 717 cm^−1^ according to the identification made in Table 2. The concentration profile (Figure 12b) shows high contributions for the third formulation (XLPE + IrgPS802) between spectra 79 to 90 corresponding to the non-irradiated third film and to the one irradiated at 67 kGy and 40 Gy/h and for spectra 103 to 117 which correspond to the 300 Gy/h irradiation dose rate. For the 40 Gy/h dose rate, from the dose 145 to 374 kGy (spectra 91 to 102) the contribution decreases with the dose and point out the Irganox PS802 consumption. For the fourth formulation (XLPE + Irg1076 + IrgPS802) high contributions are founded for spectra 118 to 126, and 142 to 156 which correspond respectively to non-irradiated sample and irradiated samples at 300 Gy/h dose rate. For the 40 Gy/h dose rate, from the dose 145 to 374 kGy (spectra 130 to 141) the contribution decreases with the dose and point out the Irganox PS802 consumption as observed for the previous sample. For the last formulation (XLPE + ATH + Irg1076 + IrgPS802) the concentration profile shows high contributions of Irganox PS802. This contribution is extremely noisy which shows a poor distribution of the additives in the sample. As previously, the Irganox PS802 consumption seems to be observed for the samples irradiated at 40 Gy/h dose rate, from the dose 145 to 374 kGy (spectra 208 to 219) with the decrease of the contribution with the dose.

The third pure spectrum (Figure 13a) corresponds to Irganox PS802 degradation products, the characteristic infrared band are found at 1735 cm^−1^, 1463 cm^−1^, 1419 cm^−1^, 1363 cm^−1^, 1311 cm^−1^, 1263 cm^−1^, 1186 cm^−1^, 1128 cm^−1^, 1056 cm^−1^, 989 cm^−1^, 788 cm^−1^ 728 cm^−1^. The concentration profile (Figure 13b) shows an important contribution for spectra 91 to 102 which correspond to the high degradation of Irganox PS802 from the dose 145 to 374 kGy at 40 Gy/h (spectra 91 to 102) shown in Figure 13. Other important contributions were shown for spectra 130 to 141 and 208 to 219 corresponding to the samples irradiated at 40 Gy/h, from the dose 145 to 374 kGy for which the Irganox PS802 consumption was also observed in Figure 13.

The fourth pure spectrum (Figure 14a) corresponds to Irganox 1076 spectrum, the characteristic infrared band are found at 1735 cm^−1^, 1463 cm^−1^, 1434 cm^−1^, 1363 cm^−1^, 1321 cm^−1^, 1164 cm^−1^, 1120 cm^−1^, 869 cm^−1^ and 717 cm^−1^. The concentration profile (Figure 14b) shows high contributions for the second formulation (XLPE + Irg1076) between spectra 40 to 48 corresponding to the non-irradiated third film and for spectra 64 to 78 which correspond to the 300 Gy/h irradiation dose rate. For the 40 Gy/h dose rate (spectra 49 to 63) the contribution decreases with the dose and points out the Irganox 1076 consumption with irradiation. For the fourth and the sixth formulation containing Irganox 1076 additive and irradiated at 40 Gy/h (spectra 127 to 141 and spectra 205 to 219 respectively), a decrease of the contribution is also observed pointing out the Irganox 1076 consumption with irradiation.

The fifth pure spectrum (Figure 15a) corresponds to oxidative species, the characteristic infrared bands are found at 1712 cm^−1^, 1463 cm^−1^, and 717 cm^−1^. The associated contributions (Figure 15b) mainly describe the degradation of the XLPE polymer. For XLPE without additives (first formulation), the contribution increases for the samples irradiated at 40 Gy/h (spectra 10 to 24) and 300 Gy/h (spectra 25 to 39) dose rates. For the dose rate at 300 Gy/h the contribution is much lower. These results are in very good agreement with the scores obtained in PCA for the XLPE and point out that the polymer is more degraded (oxidized) at low dose rate than at higher. There is no positive contribution associated with the second formulation (XLPE + Irg1076) regardless of the dose and dose rate. Therefore, it can be concluded that Irganox 1076 effectively protects the polymer against oxidative degradation at any dose and dose rate. For the third formulation (XLPE + IrgPS802), a fairly large contribution is found for the dose rate at 40 Gy/h from the dose 145 to 374 kGy (spectra 91 to 102), the contribution decreases for the dose rate at 300 Gy/h. In both cases the contribution increases with the dose. These results indicate that the Iragnox PS802 protects the polymer less effectively than Irganox 1076 against oxidative degradation. For the fourth formulation (XLPE + Irg1076 + IrgPS802) the polymer is not degraded, in this case also, the Irganox 1076 fulfilled its role. For the fifth formulation (XLPE + ATH), the profile concentration is close to that observed for the XLPE alone, which indicate that the ATH filler does not protect the polymer against oxidative degradation. Finally for the last formulation, there is no contribution of oxidative species observed due to the Irganox 1076 effect against oxidative degradation. In summary, this concentration profile of oxidative species due to the XLPE polymer degradation under irradiation clearly shows a very good efficiency of Irganox 1076 against oxidative degradation, a more limited efficiency of Irganox PS802 and an inefficiency of ATH filler.

The sixth pure spectrum (Figure 16a) corresponds to XLPE polymer, the characteristic infrared bands are found at 1463 cm^−1^ and 717 cm^−1^ according to the identification in Table 2. The contribution (Figure 16b) shows an important contribution for all spectra with a decrease of the contribution for XLPE without additives formulation and for XLPE + Irganox PS802 formulation irradiated at 40 Gy/h, which is in good agreement with the observations made previously: the samples are particularly degraded when they are irradiated at the dose rate of 40 Gy/h and in the absence of Irganox 1076 in the formulation. A decrease in the contribution is observed for the two last formulations which is consistent with the lower amount of XPLE in these formulations (67 and 65.8% vs. 99 and 98% for the other formulations).

### 3.4. Impact of Irradiation on Stabilizers Consumption Evaluated by DSC (OIT)

Consistent with the literature [7], the high antioxidant concentration in the samples results in an increase in the OIT values and, therefore, the samples containing the Irganox 1076 antioxidant show the highest OIT values. For stabilized polymers, OIT measurements can be used to give an assessment of the level of stabilization of the material, as OIT is the time required to consume all the stabilizers (especially antioxidants additives). Figure 17 shows the decrease in the OIT and therefore the consumption of antioxidants as a function of the dose of irradiation for the different samples irradiated at 40 and 300 Gy/h.

Figure 17 shows clearly that the samples without Irganox 1076 are totally oxidized from the first irradiation dose, which shows that the additives Irganox PS802 (heat stabilizer additive) and ATH (flame-retardant additive) do not have antioxidant properties. These results seem to show that Irganox PS802 is totally consumed at low doses, but previous infrared studies allow more precise interpretation and highlights that the degradation of Irganox PS802 is not complete for the high dose rate.

Irganox 1076 protects effectively against the irradiation degradation and is consumed as the dose increases.

From the OIT measurements it is not possible to give trends on the effect of dose rate on the irradiation degradation of the samples and on the effect of stabilizer consumption, as observed previously with the infrared studies.

## 4. Conclusions

Infrared spectroscopy (ATR) and DSC measurements give complementary and consistent information at different scales on radiolytic degradation of the material studied.

The radiolytic degradation is performed at room temperature on very thin samples, ensuring a homogeneous degradation (while avoiding the limitation of the oxygen diffusion). Therefore, although the infrared studies carried out by ATR-FTIR analysis are based on the first microns of the samples, the results are representative of the modification induced in the major part of the samples. This is also pointed out by the consistent results with DSC analysis, which is a more macroscopic and global measurement of the material. Some modifications at the extreme core of the material can nevertheless be impacted by crosslinking reactions, especially for high dose rate.

XLPE without any additive is degraded by gamma-irradiation. In this case, the dose rate is a determining factor in the degradation of the polymer before the dose itself, i.e., the degradation is more important at 40 Gy/h than at 300 Gy/h. Indeed, to obtain an equivalent dose, a longer exposure time is required at low dose rate which enhances the possibility of oxygen diffusion into the material and, therefore, the degradation of the material. In the 995–860 cm^−1^ spectral range cross-linking appears, and it was more present at low dose rate. The addition of 30% *w*/*w* of ATH does not seem to have a protective effect against radiolytic ageing, and this was expected because ATH is a fire retardant. The addition of 1% *w*/*w* of Irganox PS802 limits the degradation of the polymer, whereas Irganox 1076 prevents polymer degradation at any dose and dose rate. Regarding the radiolytic degradation Irganox 1076 is more effective than Irganox PS802 against oxidative degradation. From the DSC results, Irganox PS802 is totally consumed since the lowest irradiation doses. Infrared spectroscopy allows to explain its degradation mechanism. These studies show clearly, in different ways (FTIR/PCA, FTIR/SIMPLISMA, DSC) that the most effective additive against radiolytic oxidative ageing is Irganox 1076, although it can be considered that a combination of several additives could be efficient in protecting the polymer against other degradation factors besides gamma irradiation.

## Figures and Tables

**Figure 1 polymers-13-01451-f001:**
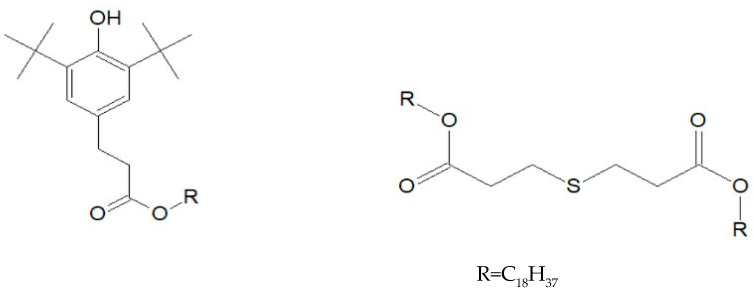
Chemical structure of Irganox 1076 (**left**) and Irganox PS802 (**right**) additives.

**Figure 2 polymers-13-01451-f002:**
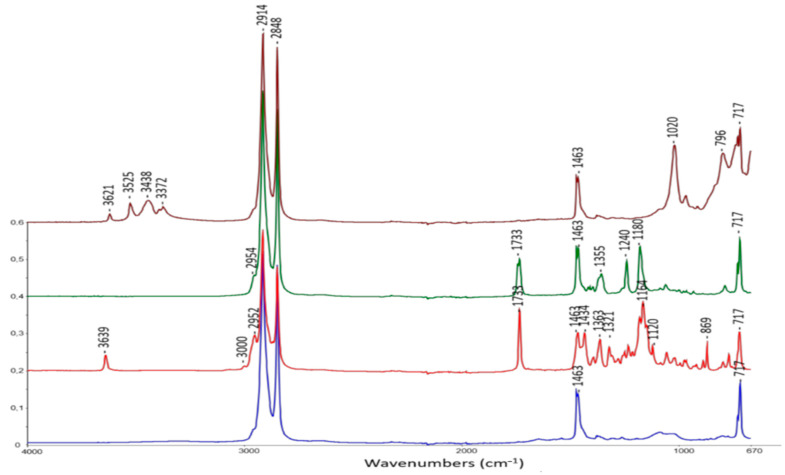
Attenuated total reflectance-Fourier transform infrared (ATR-FTIR) spectra of unaged materials: crosslinked polyethylene (XLPE, blue), XLPE + Irganox 1076 (red), XLPE + Irganox PS802 (green) and XLPE + alumina trihydrate (ATH, brown).

**Figure 3 polymers-13-01451-f003:**
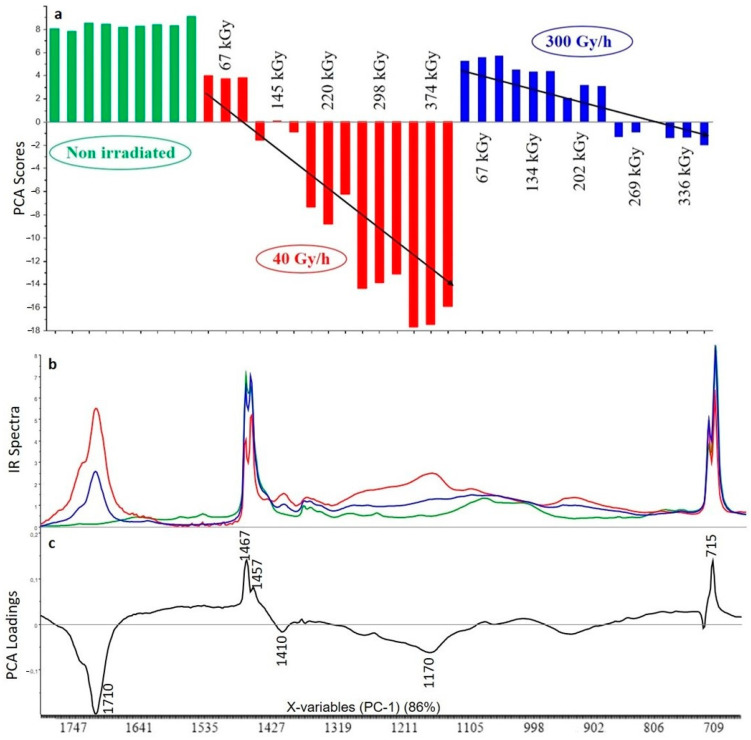
Principal component analysis (PCA) scores (**a**), FTIR spectra for highest dose (**b**) and PCA loadings (**c**) of the first principal component (PC1) for the analysis of XLPE polymer without additive (green: unaged samples, red: samples irradiated at 40 Gy/h, blue: samples irradiated at 300 Gy/h).

**Figure 4 polymers-13-01451-f004:**
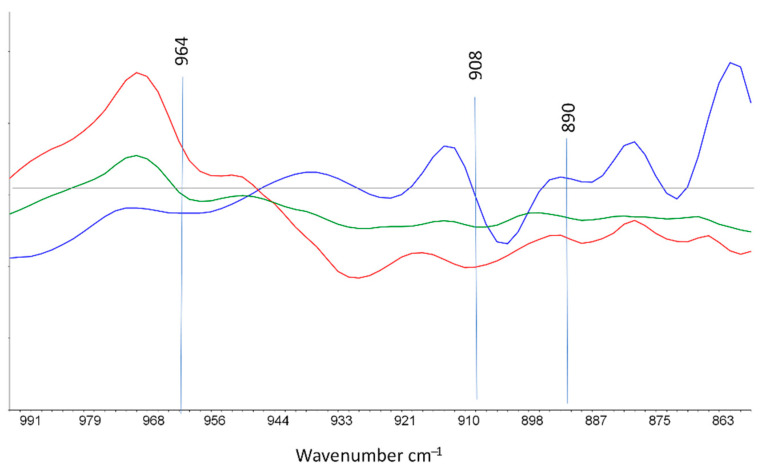
First derivative FTIR spectra in the range 995–860 cm^−1^ of XLPE irradiated at 374 kGy and 40 Gy/h (red), XLPE irradiated at 336 kGy and 300 Gy/h (green), and non-irradiated XLPE (blue).

**Figure 5 polymers-13-01451-f005:**
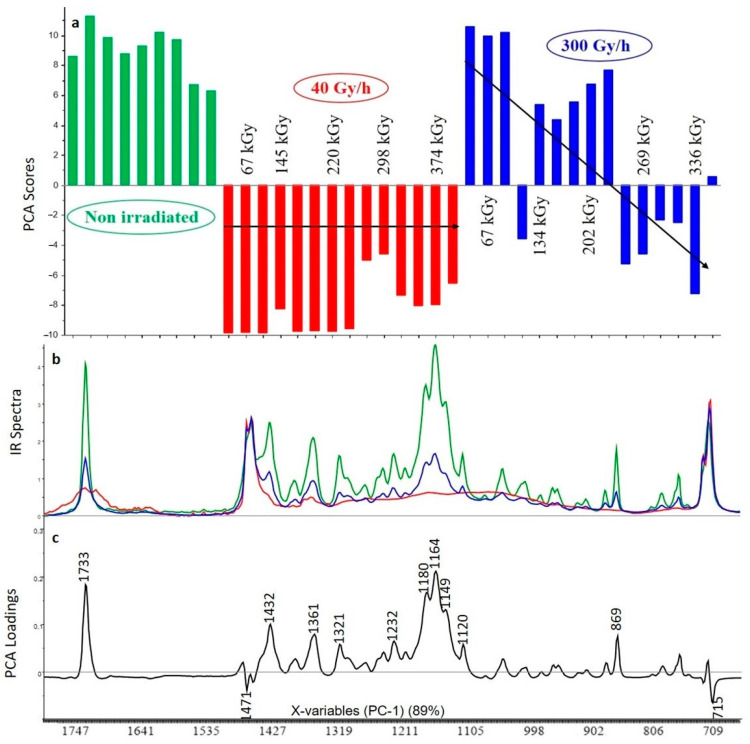
PCA scores (**a**), FTIR spectra for highest dose (**b**) and PCA loadings (**c**) of the first principal component (PC1) for the analysis of XLPE with Irganox 1076 (green: unaged samples, red: samples irradiated at 40 Gy/h, blue: samples irradiated at 300 Gy/h).

**Figure 6 polymers-13-01451-f006:**
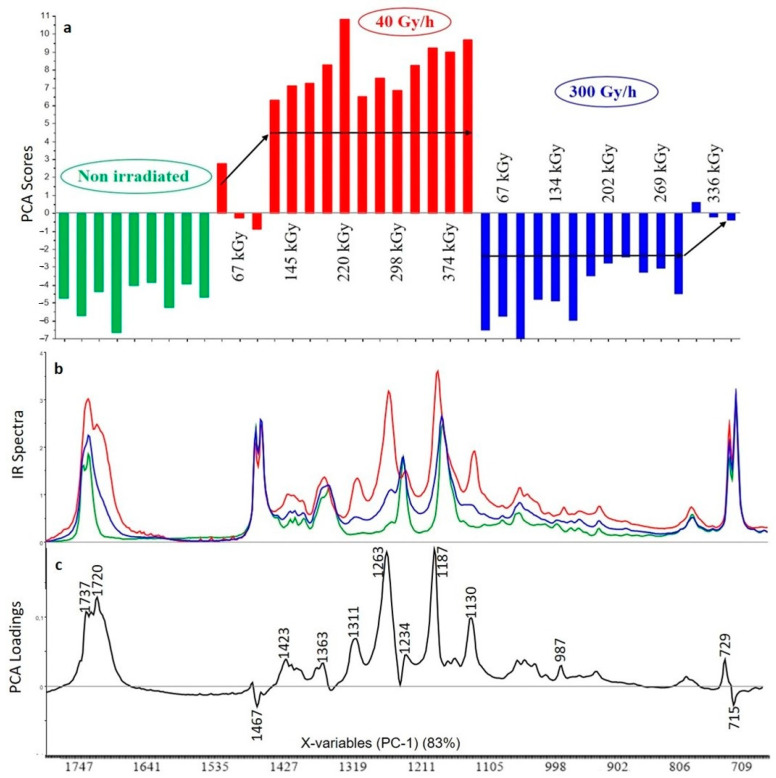
PCA scores (**a**), FTIR spectra for highest dose (**b**) and PCA loadings (**c**) of the first principal component (PC1) for the analysis of XLPE with Irganox PS802 (green: unaged samples, red: samples irradiated at 40 Gy/h, blue: samples irradiated at 300 Gy/h).

**Figure 7 polymers-13-01451-f007:**
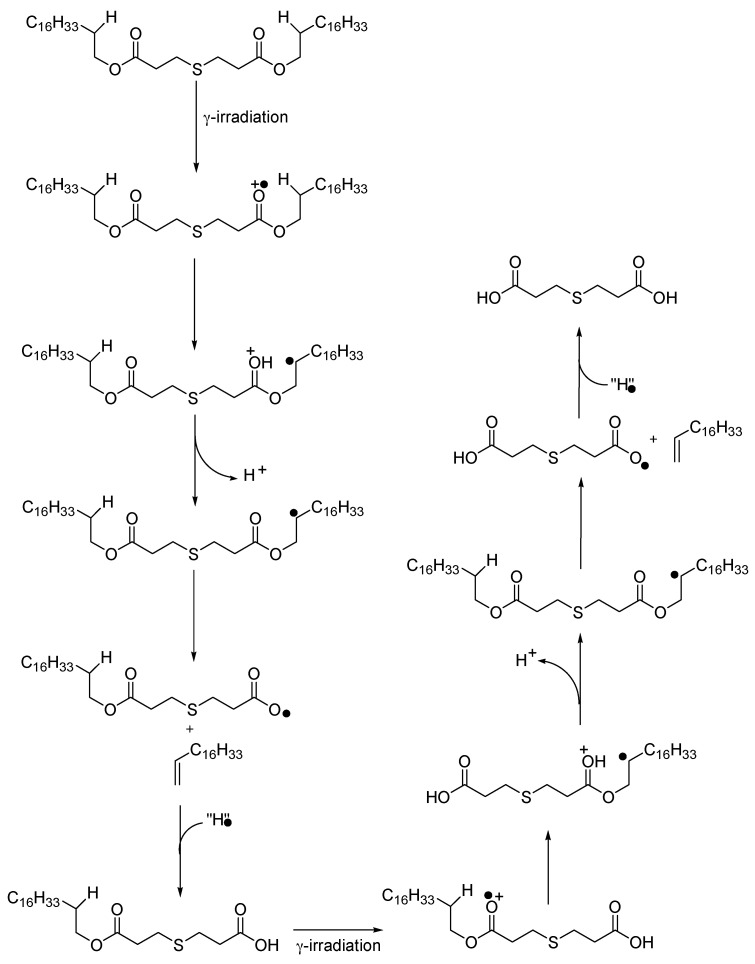
Proposed pathway for the degradation of Irganox PS802 into 3,3′-thiodipropionic acid.

**Figure 8 polymers-13-01451-f008:**
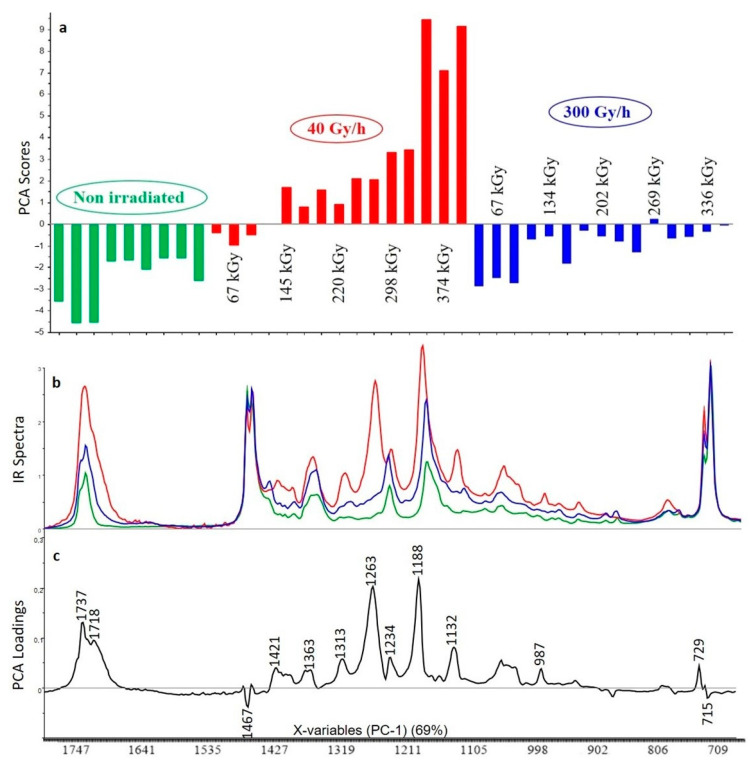
PCA scores (**a**), FTIR spectra for highest dose (**b**) and PCA loadings (**c**) of the first principal component (PC1) for the analysis of XLPE with Irganox 1076 and Irganox PS802 (green: unaged samples, red: samples irradiated at 40 Gy/h, blue: samples irradiated at 300 Gy/h).

**Figure 9 polymers-13-01451-f009:**
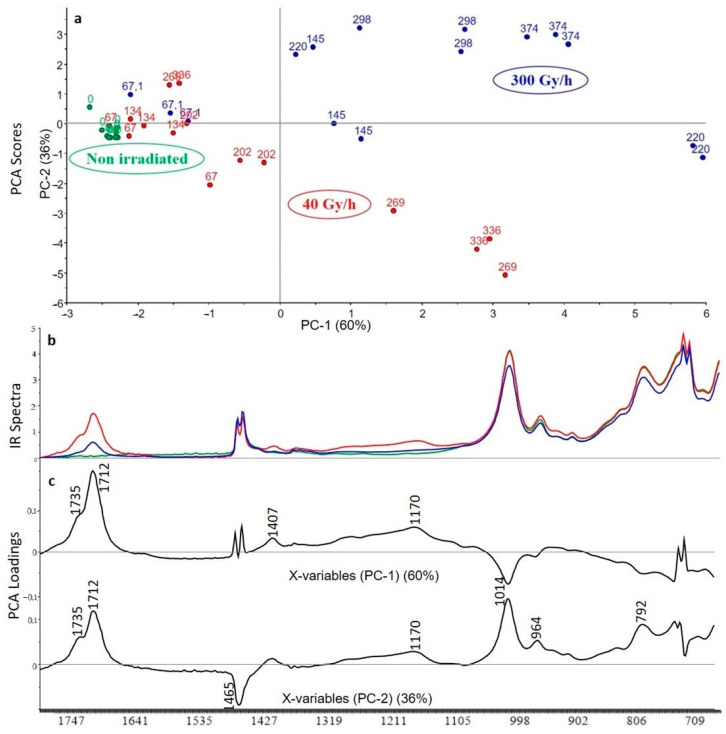
PCA scores (**a**), FTIR spectra for highest dose (**b)** and PCA loadings (**c**) of the first and second principal components (PC1 and PC2) for the analysis of XLPE with ATH (green: unaged samples, red: samples irradiated at 40 Gy/h, blue: samples irradiated at 300 Gy/h).

**Figure 10 polymers-13-01451-f010:**
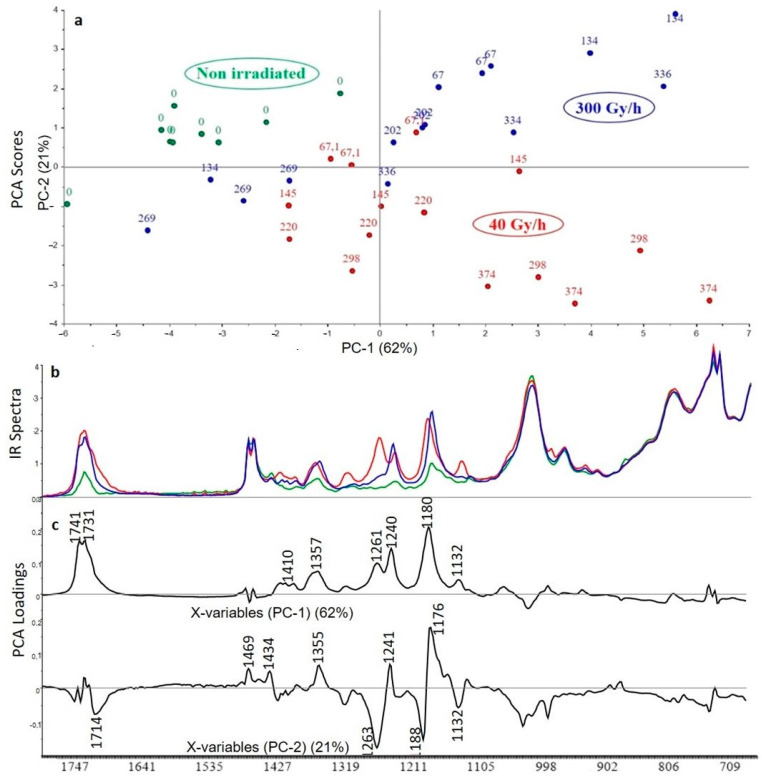
PCA scores (**a**), FTIR spectra for highest dose (**b**) and PCA loadings (**c**) of the first and second principal components (PC1 and PC2) for the analysis of XLPE with ATH, Irganox 1076 and Irganox PS802 (green: unaged samples, red: samples irradiated at 40 Gy/h, blue: samples irradiated at 300 Gy/h).

**Figure 11 polymers-13-01451-f011:**
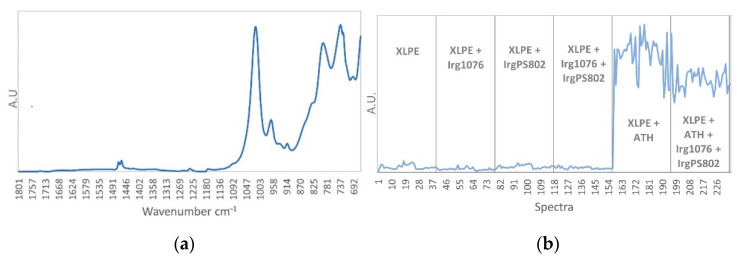
(**a**) ATH pure spectrum. (**b**) ATH concentration profile.

**Figure 12 polymers-13-01451-f012:**
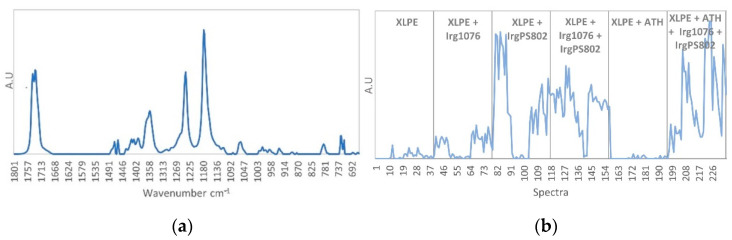
(**a**) IrgPS802 pure spectrum. (**b**) IrgPS802 concentration profile.

**Figure 13 polymers-13-01451-f013:**
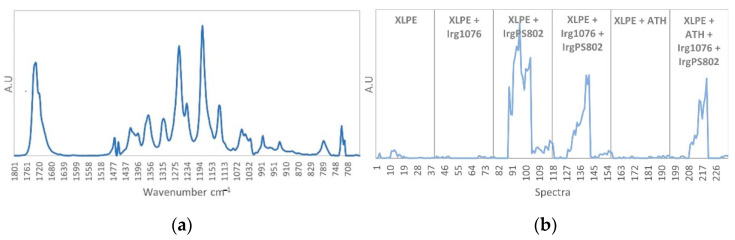
(**a**) IrgPS802 degradation products pure spectrum. (**b**) IrgPS802 degradation products concentration profile.

**Figure 14 polymers-13-01451-f014:**
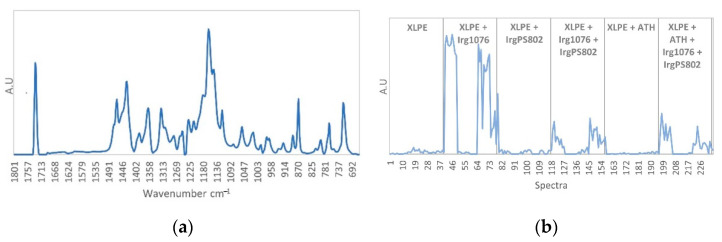
(**a**) Irg1076 pure spectrum. (**b**) Irg1076 concentration profile.

**Figure 15 polymers-13-01451-f015:**
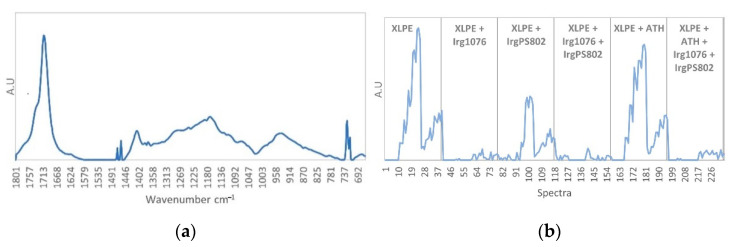
(**a**) Oxidative species pure spectrum. (**b**) Oxidative species concentration profile.

**Figure 16 polymers-13-01451-f016:**
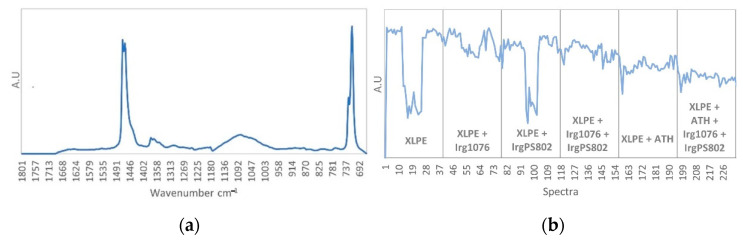
(**a**) XLPE pure spectrum. (**b**) XLPE concentration profile.

**Figure 17 polymers-13-01451-f017:**
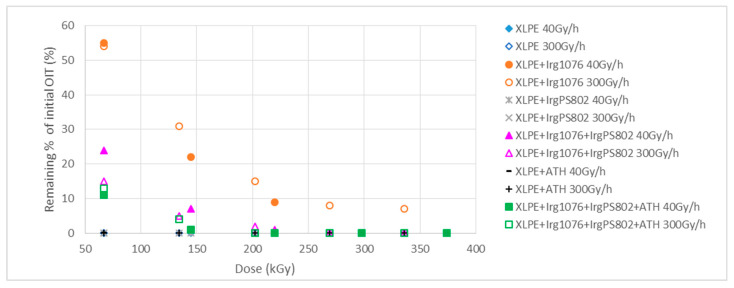
Remaining percentages of initial oxidative induction time (OIT) for the different samples (full markers: samples irradiated at 40 Gy/h, empty markers: samples irradiated at 300 Gy/h).

**Table 1 polymers-13-01451-t001:** Integrated gamma doses (and ageing time) for the five withdrawals at each dose rate (RA(RT) 40: radiolytic ageing of 40 Gy/h at room temperature, RA(RT)300: radiolytic ageing of 300 Gy/h at room temperature).

Withdrawal	RA(RT)40	RA(RT)300
1	67.1 kGy (42 days)	67.0 kGy (7 days)
2	145.0 kGy (84 days)	134.0 kGy (14 days)
3	220.0 kGy (125 days)	202.0 kGy (21 days)
4	298.0 kGy (168 days)	269.0 kGy (28 days)
5	374.0 kGy (210 days)	336.0 kGy (35 days)

**Table 2 polymers-13-01451-t002:** Identification of the main FTIR bands of the unaged materials.

	Functional Group	Type of Vibration	XLPE	XLPE + Irganox 1076	XLPE + Irganox PS802	XLPE + ATH
3639	ν–O–H phenol	Stretching		x		
3621-3372	ν–Al–OH	O–H stretching				x
3000	ν=C–H aromatic	Stretching		x		
2954	ν–CH_2_–S	CH_2_ antisymmetric stretching			x	
2952	ν–CH_3_	Stretching		x		
2914-2848	ν–CH_2_–	Antisymmetric and Symmetric stretching	x	x	x	x
1743-1733	ν–C=O ester	Stretching		x	x	
1473-1463	δ–CH_2_–	Deformation in the plane	x	x	x	x
1434-1321	δ–O–H, ν C–O	Combination		x		
1355	ω–CH_2_–S–	Wagging			x	
1300-1000	ν–C–O ester	Stretching		x	x	
1240-1180	ν–C–O	Asymmetric stretching			x	
1175-1150	δ–O–H, ν C–O	Combination		x		
1120	δ–C–H aromatic	Deformation		x		
1020-698	ν–Al–O	Stretching				x
869	δ=C–H aromatic	Bending		x		
728-717	δ–CH_2_–	Inner rocking vibration of CH_2_ in the crystalline part	x	x	x	x

**Table 3 polymers-13-01451-t003:** Correspondence of formulations and spectra.

Formulations	Irradiation Conditions	Spectra
XLPE without additive	Non irradiated	1 to 9
	40 Gy/h: 67 to 374 kGy	10 to 24
	300 Gy/h: 67 kGy to 336 kGy	25 to 39
XLPE + Irg1076 (99% + 1% *w*/*w*)	Non irradiated	40 to 48
	40 Gy/h: 67 to 374 kGy	49 to 63
	300 Gy/h: 67 kGy to 336 kGy	64 to 78
XLPE + IrgPS802 (99% + 1% *w*/*w*)	Non irradiated	79 to 87
	40 Gy/h: 67 to 374 kGy	88 to 102
	300 Gy/h: 67 kGy to 336 kGy	103 to 117
XLPE + Irg1076 + IrgPS802 (98% + 1% + 1% *w*/*w*)	Non irradiated	118 to 126
	40 Gy/h: 67 to 374 kGy	127 to 141
	300 Gy/h: 67 kGy to 336 kGy	142 to 156
XLPE + ATH (67% + 33% *w*/*w*)	Non irradiated	157 to 165
	40 Gy/h: 67 to 374 kGy	166 to 180
	300 Gy/h: 67 kGy to 336 kGy	181 to 195
XLPE + ATH + Irg1076 + IrgPS802 (65.8% + 32.8% + 0.7% + 0.7% *w*/*w*)	Non irradiated	196 to 204
	40 Gy/h: 67 to 374 kGy	205 to 219
	300 Gy/h: 67 kGy to 336 kGy	220 to 234

## Data Availability

The data are available at the end of the project at https://www.team-cables.eu/, accessed on 1 March 2021.

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
