# Peer review of "Influence of Gamma Irradiation on Electric Cables Models: Study of Additive Effects by Mid-Infrared Spectroscopy"

_polymers, 2021, doi:10.3390/polym13091451_

Round 1

Reviewer 1 Report

The article is interesting but I miss the reference to the strength properties.

Author Response

Response

We have added references to better support our work. In addition, we worked on a curve resolution method to extract the spectra of products involved in aging as well as their concentration profiles. This has improved the publication

Reviewer 2 Report

Review of polymers-1148151 “Influence of gamma irradiation on electric cables models: Study of additive effects by mid infrared spectroscopy”

Wherever nuclear power plants are in use, their safe operation is a key issue and one aspect of this is the durability of the cables over the operation time of the plant. Thus, the investigation of degradation processes of cable insulation materials under influence of gamma-irradiation, oxidative atmosphere (and heat) is important. In the present study, the degradation of the widely used insulation material XLPE was studied, either in its pure form or in combination with additives (two antioxidants of the Irganox series, alumina trihydrate or combinations thereof). While the choice of samples and the experimental procedure of the study (OIT measurements and ATR-FTIR in dependence of gamma dose and dose rates) is convincing, the manuscript has many weak points.

  1. The authors have chosen a topic that has been thoroughly studied in the recent past. I find the citation of previous publications is largely lacking. Relevant examples that have not been discussed in the manuscript include:
  • DOI: 10.1109/ICD46958.2020.9341895, Suraci et al., “Chemical and electrical characterization of XLPE cables exposed to radio-thermal aging”
  • org/10.1016/j.polymdegradstab.2018.09.011, Bowler et al., “Quantitative analysis of changes in antioxidant in crosslinked polyethylene (XLPE) cable insulation material exposed to heat and gamma radiation”
  • DOI:10.1002/tee.23023, Ohki et al., “Comparison of the Effects of Heat and Gamma Irradiation on the Degradation of Cross-Linked Polyethylene”
  • org/10.1016/S0969-806X(99)00328-X, Singh, “Irradiation of polyethylene: Some aspects of crosslinking and oxidative degradation”
  • org/10.1016/j.radphyschem.2007.02.101, Ghaffari and Agmadian, “Investigation of antioxidant and electron beam radiation effects on the thermal oxidation stability of low-density polyethylene”
  • DOI: 10.1016/S0141-3910(01)00012-X, Suarez and Mano, “Characterization of degradation on gamma-irradiated recycled polyethylene blends by scanning electron microscopy”

Especially the first three examples are very close to the present study and the additional benefit of reading the current publication should be discussed in the introduction. In detail, the study of Suraci et al. uses the same characterization methods, uses presumably a very comparable polymer material (“This is made up of silane crosslinked polyethylene stabilized with 1 phr of primary AO (phenol-based) and 1 phr of secondary AO (thioether-based).”) and it uses additionally a fabricated cable as a specimen, not a polymer sheet. The authors are highly encouraged to deal with the listed publications and use their findings in the introduction as well as in the discussion of the results.

  1. Let us start with a positive remark – the detailed assignment of vibrational bands to functional groups is a strength of the manuscript, since that is seldomly done in comparable studies. This strength should be developed by distinction between crystalline and amorphous parts of XLPE if reference data for these domains are available. Possibly, this could lead to a detailed description of what happens during aging, especially because Singh describes that additional crosslinking may happen due to gamma irradiation.

  1. A key question stays unanswered throughout the whole manuscript: Does the stabilizer prevent the polymer from oxidation in expense of its own consumption or are both (polymer and stabilizer) affected? Following points may help to tackle this question.

  1. The use of the SNV normalization has been performed, but its necessity has not been discussed in the paper. Through normalization, the information about the actual concentration of the present species gets lost and it would be interesting whether the intensity of undegraded polymer bands stays constant or the polymer is consumed.
  2. The PCA method applied on the different experimental series do not seem to produce a benefit to interpret the IR spectra, because the varying PCA scores only show that the spectra are changing with increasing dose applied, but the PCA loading is not distinctively connected to one component present in the sample. Subsequently, the presentation of PCA scores and loadings distracts the reader from the actual results.
  3. Thus, the manuscript can be improved if the reader is confronted with the time-dependent spectra of each measurement series and increasing or decreasing signals are discussed. Through comparison of the spectra shown in Figure 3 with spectra shown in Figure 4, 5, 7, 8 or 9 it could be possible to distinguish qualitatively what has happened during degradation of the samples with additives. Additionally, it should be possible to separate the spectral components of polyethylene, the two antioxidants, aluminium trihydrate and the oxidation products of polyethylene or the oxidized antioxidants. This can be eventually done by a multivariate data analysis using PCA or PLS, alternatively a so-called complemental hard modelling may be useful for this problem.

  1. Furthermore, the quality of the figures containing spectra should be improved by magnifying the axes description and there are several typos in the manuscript that require a spell check (e.g. 300 °C temperature in line 208).

In conclusion, the manuscript may be suitable for publication, following major revisions. The authors should answer the points addressed before and substantially improve the data analysis and discussion of the IR spectra.

Author Response

  1. The authors have chosen a topic that has been thoroughly studied in the recent past. I find the citation of previous publications is largely lacking. Relevant examples that have not been discussed in the manuscript include:
  • DOI: 10.1109/ICD46958.2020.9341895, Suraci et al., “Chemical and electrical characterization of XLPE cables exposed to radio-thermal aging”
  • org/10.1016/j.polymdegradstab.2018.09.011, Bowler et al., “Quantitative analysis of changes in antioxidant in crosslinked polyethylene (XLPE) cable insulation material exposed to heat and gamma radiation”
  • DOI:10.1002/tee.23023, Ohki et al., “Comparison of the Effects of Heat and Gamma Irradiation on the Degradation of Cross-Linked Polyethylene”
  • org/10.1016/S0969-806X(99)00328-X, Singh, “Irradiation of polyethylene: Some aspects of crosslinking and oxidative degradation”
  • org/10.1016/j.radphyschem.2007.02.101, Ghaffari and Agmadian, “Investigation of antioxidant and electron beam radiation effects on the thermal oxidation stability of low-density polyethylene”

These articles were added and discussed.

  • DOI: 10.1016/S0141-3910(01)00012-X, Suarez and Mano, “Characterization of degradation on gamma-irradiated recycled polyethylene blends by scanning electron microscopy”

Especially the first three examples are very close to the present study and the additional benefit of reading the current publication should be discussed in the introduction. In detail, the study of Suraci et al. uses the same characterization methods, uses presumably a very comparable polymer material (“This is made up of silane crosslinked polyethylene stabilized with 1 phr of primary AO (phenol-based) and 1 phr of secondary AO (thioether-based).”) and it uses additionally a fabricated cable as a specimen, not a polymer sheet. The authors are highly encouraged to deal with the listed publications and use their findings in the introduction as well as in the discussion of the results.

We have included the references in the introduction and in our discussion. Our study is part of the Team Cables project like the Suraci reference, but we work under different conditions since we do not have thermal aging for our samples.

  1. Let us start with a positive remark – the detailed assignment of vibrational bands to functional groups is a strength of the manuscript, since that is seldomly done in comparable studies. This strength should be developed by distinction between crystalline and amorphous parts of XLPE if reference data for these domains are available. Possibly, this could lead to a detailed description of what happens during aging, especially because Singh describes that additional crosslinking may happen due to gamma irradiation.

For the polymer without additive, we have shown cross linking. This is not possible in the presence of additives because their spectra overlapped in the spectral region 980-850 cm-1 .

  1. A key question stays unanswered throughout the whole manuscript: Does the stabilizer prevent the polymer from oxidation in expense of its own consumption or are both (polymer and stabilizer) affected? Following points may help to tackle this question.

Discussion was added in the manuscript.

  1. The use of the SNV normalization has been performed, but its necessity has not been discussed in the paper. Through normalization, the information about the actual concentration of the present species gets lost and it would be interesting whether the intensity of undegraded polymer bands stays constant or the polymer is consumed.

Explanation of SNV is added in material and method part.

  1. The PCA method applied on the different experimental series do not seem to produce a benefit to interpret the IR spectra, because the varying PCA scores only show that the spectra are changing with increasing dose applied, but the PCA loading is not distinctively connected to one component present in the sample. Subsequently, the presentation of PCA scores and loadings distracts the reader from the actual results.

PCA analysis allows to interpret effect of dose and dose rate for each additive.

  1. Thus, the manuscript can be improved if the reader is confronted with the time-dependent spectra of each measurement series and increasing or decreasing signals are discussed. Through comparison of the spectra shown in Figure 3 with spectra shown in Figure 4, 5, 7, 8 or 9 it could be possible to distinguish qualitatively what has happened during degradation of the samples with additives. Additionally, it should be possible to separate the spectral components of polyethylene, the two antioxidants, aluminium trihydrate and the oxidation products of polyethylene or the oxidized antioxidants. This can be eventually done by a multivariate data analysis using PCA or PLS, alternatively a so-called complemental hard modelling may be useful for this problem.

A curve resolution study has been added, which effectively allows discussion of the relative effects of each additive.

  1. Furthermore, the quality of the figures containing spectra should be improved by magnifying the axes description and there are several typos in the manuscript that require a spell check (e.g. 300 °C temperature in line 208).

The quality of figures was improved.

Concerning line 208, unity noted in the manuscript is correct, we are not talking about temperature, but dose rate

Reviewer 3 Report

The manuscript entitled "Influence of gamma irradiation on electric cables models: Study of additive effects by mid infrared spectroscopy" reports about a study dealing with the spectroscopical characterization of different XLPE-based materials (containing commercial antioxidants and/or flame retardants) subjected to gamma irradiation. 

In my view, the major concern of the manuscript is the lack of originality with respect to several works already published in the scientific literature dealing with the influence of gamma irradiation on polyolefin-based materials used for the formulation of insulation cables. In fact, it is not clear which is the updating as compared to what already published on the same topic, given that both the used polymer matrix and the additives are commonly used commercial materials. 

Author Response

Response

We have added references to better support our work. In addition, we worked on a curve resolution method to extract the spectra of products involved in aging as well as their concentration profiles. This has improved the manuscript and added an approach to a better understanding of additives behaviour into polymers.

Reviewer 4 Report

The manuscript Influence of gamma irradiation on electric cables models: 2Study of additive effects by mid infrared spectroscopy  is interestingly written, well corrected. In my opinion, it should have been published. However a minor editorial correction is required:

Line 392-408

Line 422-435

Author Response

Text has been changed to make the discussion clearer

Round 2

Reviewer 2 Report

In comparison to the previous version, sufficient changes have been made to improve the manuscript. Most notably, they deconvoluted the mixture spectra using the SIMPLISMA algorithm what allowed to distinguish whether the polymer or the antioxidants are affected by oxidation processes. This allows to estimate that Irganox 1076 is a superior antioxidant in comparison with Irganox PS802. Also, the revised introduction and the attempt to investigate crosslinking improve the manuscript.

Again, the spelling needs to be checked, especially in the newly added parts. The greek letters have not been displayed. Figure 1 may be improved by a higher resolution, figure 14a shall be revised.

Reviewer 3 Report

Although the manuscript has been intensively revised by the Authors, my opinion about the poor novelty of the submitted work remains unchanged. The analyses added in the revised version are interesting and well commented, but in my view the work is very similar to other  studies already present in the scientific literature and it not deserves to be published on Polymers.